# Latent Debate: A Surrogate Framework for Interpreting LLM Thinking

## Abstract

Understanding the internal thinking process of Large Language Models (LLMs) and the cause of hallucinations remains a key challenge. To this end, we introduce *latent debate*, a novel framework for interpreting model predictions through the lens of implicit internal arguments. Unlike the current work of self-consistency and multi-agent debate, which relies on explicit debates among multiple answers or multiple models, latent debate captures the hidden supporting and attacking signals that arise within a single model during a single inference. We first present a model- and task-agnostic conceptual framework, and then instantiate it symbolically to approximate the thinking process of LLMs on True/False prediction tasks. Empirical studies demonstrate that latent debate is a faithful structured surrogate model that has highly consistent predictions with the original LLM. Beyond interpretability, we demonstrate that latent debate provides a strong baseline for hallucination detection. Further analysis reveals strong correlations between hallucinations and debate patterns, such as a high degree of latent debates in the middle layers is linked to a higher risk of hallucinations. These findings position latent debate as a potential framework for understanding internal mechanisms of LLMs, especially for scenarios where internal (dis)agreements appear during the inference steps.

## 1 Introduction

Large Language Models (LLMs) have made remarkable progress on many reasoning tasks, yet they continue to suffer from hallucinations (Xu et al., 2024b; Huang et al., 2025). For example, LLMs may generate answers that contradict user prompts and or conflict with the source of training data (Ji et al., 2023; Kalai et al., 2025; Bang et al., 2025), seriously undermining their reliability and trustworthiness. This is further aggravated by the fact that, due to their opacity, it is difficult to understand why LLMs make given predictions, or why their "thinking" process is flawed.

Recent work in mechanistic interpretability has examined hallucinations through various internal signals, including activations (Ferrando et al., 2025), attention patterns (Chuang et al., 2024), and hidden states (Azaria & Mitchell, 2023a). Another line of relevant research (Wang et al., 2022) leverages external consistency, i.e., the agreement among multiple answers, to analyze hallucination behaviors. Their findings reveal that hallucinated outputs tend to have low self-consistency (Wang et al., 2022). This phenomenon suggests that strong agreement among multiple answers often yields more certain and accurate answers, whereas disagreement indicates higher uncertainty and can serve as a good signal for understanding hallucinations. Subsequent approaches (Irving et al., 2018; Du et al., 2024; Chen et al., 2024b; Liang et al., 2024a) further introduce Multiple-Agent Debate (MAD) to reduce hallucinated answers via a debate process of multiple language models, often outperforming single-model baselines.

Inspired by these prior studies of mechanistic interpretability and disagreement/debate, we aim to understand how hallucinations emerge within a model but shifting to *latent debate*, i.e., arising among different layers and "thinking" steps within an individual model and a single inference (Chuang et al., 2023; Liang et al., 2024b; Xie et al., 2024), rather than externally to it as in prior work. Unlike in a conventional, human debate, where arguments are in natural language, the arguments in our latent debate are to be understood in a metaphorical sense as they correspond to latent states in models. Intuitively, latent debates aggregate metaphorical arguments to reflect the thinking and decision-making process beneath the surface. In psychological theories, the human thinking process

often involves internal debate-like behaviors such as inner speech (Barker & Wiseman, 1966) and the dialogical self (Hermans, 2001). Here, we extend this psychological insight to models introducing latent debates to describe an analogous process taking place within a model.

We focus on two key research questions in this work: (1) *Can we use latent debate to model the LLM thinking process?* (2) *Can latent debate identify hallucinations?*

To answer the first question, we present a conceptual framework of latent debate that depicts (dis)agreement within a model, which is model- and task-agnostic. The framework consists of three abstract components: latent arguments derived from internal signals, an argument interpreter that translates these implicit arguments into human-readable opinions such as supporting or attacking a claim, and a thinking module that aggregates them to make the final decisions. We then instantiate this framework in decoder-based LLMs on True/False prediction tasks (see case studies in Figure 1), where hidden states serve as latent arguments, the unembedding matrix acts as the argument interpreter, and the thinking module is realized through a symbolic argumentation framework, in the spirit of (Čyras et al., 2021). An empirical study demonstrates that this latent debate acts as a structured surrogate model, providing a faithful approximation of LLM thinking, which achieves up to 98.3% consistency with LLaMA-13B decisions. These findings validate that our latent debate can imitate the thinking process of LLM true/false tasks.

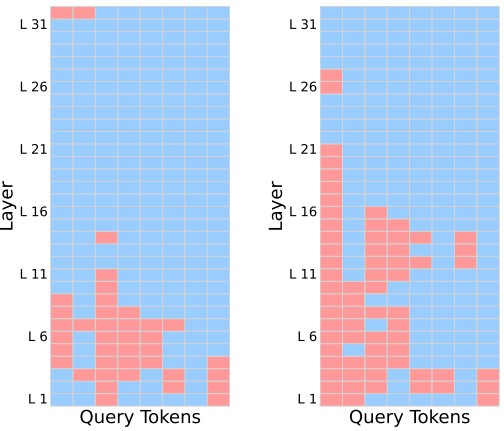

(a) **Correct**: *The city of Zhangzhou is in China.*

(b) **Hallucinated:** *The letter J is the most commonly used letter in English-language writing.*

Figure 1: Visualizations of our latent debate for two claims (We use the last few tokens of Llama-8B). Red cells represent attacking arguments, while blue cells represent supporting arguments. More controversy leads to hallucination.

To address the second question, we extract features from the latent debate graph, e.g., the number of internal debates and argument strengths, and train a simple MLP classifier to distinguish hallucinated from non-hallucinated outputs. We find that our latent debate can achieve highly competitive performance in hallucination detection. We then use SHAP attribution scores (Lundberg & Lee, 2017a) to identify which features most strongly drive hallucination predictions. Our analyses indicate that a high degree of latent debate, particularly in the middle layers, is the strongest predictor of hallucination.

In summary, our contributions are threefold. (1) We propose latent debate, a novel, model-agnostic framework that leverages internal arguments to interpret a model's thinking process. (2) We present a symbolic instantiation of latent debate that serves as a faithful surrogate for LLM True/False tasks. (3) We develop a debate-based MLP to detect hallucinations, which help us identify distinct debate patterns, especially intense internal debates in middle layers, that correlate strongly with hallucination in LLMs.

## 2 RELATED WORK

### 2.1 MULTIPLE AGENT DEBATE

Multiple-agent debate (MAD) has emerged as a powerful approach for improving factuality and reasoning. Pioneering work on AI safety via debate (Irving et al., 2018) models debate as a self-play game with a (human) judge and provides core theoretical motivation. Recent work adopts multiple language model agents to debate over individual responses jointly, with the final decision made either through consensus (Du et al., 2024; Chen et al., 2024b) or by a judge (Liang et al., 2024a). This debate strategy can outperform single model baselines on a wide variety of reasoning tasks.

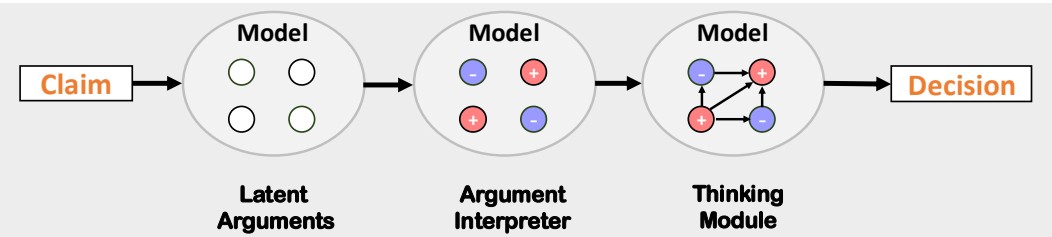

Figure 2: The overall framework of latent debate. Given an input *claim*, our method generates a set of *latent arguments*, i.e., model components (raw latent signals) that convey the model's opinions toward the claim. These arguments are then processed by the *argument interpreter*, identifying the arguments' supporting or attacking stance towards the claim. The resulting attacking and supporting arguments are fed into the *thinking module*, which applies a procedure to reach the final *decision*.

Subsequent research proposes refined debate approaches (Li et al., 2024; Liu et al., 2024) using MAD as an evaluator (Chan et al., 2024). In this work, we focus on debates operating implicitly within a model or agent rather than externally visible debates among multiple agents.

## 2.2 INTERNAL CONSISTENCY

Another line of work focuses on how to use the consistency of internal model states, such as logits and activations, to improve model outputs (Liang et al., 2024b). For example, DoLa (Chuang et al., 2023) proposes a decoding strategy that contrasts logits between later layers and earlier layers for improving LLM truthfulness. Xie et al. (2024) adopts internal consistency, i.e., how middle layers' predictions (dis)agree with the final layer, to guide LLM decoding. In this work, we aim to use latent (dis)agreements to build a surrogate framework for interpreting the thinking process of a model rather than enhancing model outputs.

## 2.3 COMPUTATIONAL ARGUMENTATION IN EXPLAINABLE AI

*Argumentation Frameworks (AFs)* (Dung, 1995), are a fundamental formalism in *computational argumentation* (Atkinson et al., 2017; Čyras et al., 2021), a well-established research area in AI. According to (Dung, 1995), an AF consists of a set of *arguments* and a binary *attack* relation among them. Arguments are seen as abstract entities, while the attack relation captures conflicts between arguments. AFs and their extensions, such as incorporating weights for arguments and a support relation between arguments, collectively referred to as Quantitative Bipolar Argumentation Frameworks (QBAFs) (Rago et al., 2016; Baroni et al., 2019; Čyras et al., 2021), have been widely adopted in Explainable AI (XAI). AFs can serve as *surrogate models* to approximate the inner structure and decision-making process of AI systems (Čyras et al., 2021; Potyka, 2021; Potyka et al., 2023; Ayoobi et al., 2023). Beyond serving as surrogates, AFs can also be explicitly integrated into AI or LLM systems to enhance explainability(Freedman et al., 2025; Čyras et al., 2021; Vassiliades et al., 2021; Engelmann et al., 2022; Guo et al., 2023). In this work, we are the first to introduce the model-agnostic concept of latent debate and adopt an AF as the thinking module of a model.

## 3 METHODOLOGY

### 3.1 PROBLEM STATEMENT

In this work, we aim to obtain a *structured surrogate model* to understand the internal 'thinking' of a target model, as opposed to a *conventional surrogate model* solely imitating the input-output behaviour of the model. More specifically, given a target model $M : \mathcal{X} \to \mathcal{Y}$, a conventional surrogate model $S : \mathcal{X} \to \mathcal{Y}$ imitates the input-output behaviour of the target model, i.e., $S(x) \approx M(x)$. Such surrogates offer a simplified and interpretable approximation of the model's outputs (Asher et al., 2015; Kudela & Matousek, 2022). This type of surrogate does not faithfully reflect the thinking (or decision-making) process of the model. To address this limitation, structured surrogate models explicitly accounts for the model's internal organization (Munk et al., 2022; Páez, 2024). Supposing

the target model has a known structure:

$$M : \mathcal{X} \to \mathcal{H} \times \mathcal{Y}, \qquad M(x) = (h(x), y(x)) \tag{1}$$

where $h(x) \in \mathcal{H}$ is the internal structure, our goal is to construct:

$$S(x) = (\hat{h}(x), \hat{y}(x)) \tag{2}$$

that approximates the internal computational structure of the target model, i.e., $\hat{h}(x) \approx h(x)$, while, at the same time, being faithful to the target model by making highly consistent predictions with it, i.e., $\hat{y}(x) \approx y(x)$.

To obtain structured surrogate models $S(x)$ with the above characteristics, we define the concept of *latent debate* as follows.

## 3.2 CONCEPTUAL FRAMEWORK

**Latent Debate.** A latent debate is an internal, implicit form of argumentation that happens within a single model (or agent). Instead of having multiple explicit agents participating in a debate, latent debate refers to the hidden inconsistency inside the model that simultaneously carries supporting and attacking arguments toward a claim. These arguments are not directly expressed in natural language but shape the model's thinking process beneath the surface. The strength between supporters and attackers may be imbalanced. Overwhelming supporters can lead to a very certain positive decision, and vice versa. This uncertainty reflects how the model arrives at a final decision. The latent debate consists of three key components: latent arguments, argument interpreter, and thinking module, as shown in Figure 2.

**Latent Arguments.** A latent argument refers to an internal signal within a model that implicitly conveys supporting or attacking opinions toward a claim. Such signals can derive from different sources, like activations or attention patterns. Because they live in the model's latent space, these arguments are not directly visible or human-readable, but they still express how intermediate steps "think" about the claim.

**Argument Interpreter.** The argument interpreter is the tool that makes these latent arguments interpretable. It translates latent arguments into a form we can understand, such as a binary label. At the same time, it tells us how strongly each argument supports or attacks the claim, turning vague internal signals into measurable opinions.

**Thinking Module.** The thinking module combines all the decoded arguments to reach a final decision. It looks at how the arguments interact — some supporting, some attacking — and weighs them against each other. By aggregating these arguments and how they interact, the module produces a final outcome that reflects the overall internal debate of the model.

It is important to note that this framework is not tied to any specific model architecture. The notions of latent arguments, argument interpreter, and thinking module are abstract components that can be realized in many different ways. For example, latent arguments may be instantiated through hidden states, attention patterns, or other internal signals; argument interpreters can be designed using projection, probing, or alternative interpretability tools; and thinking modules may adopt symbolic argumentation frameworks, probabilistic aggregation, machine-learning methods, including artificial neural networks. This flexibility ensures that the latent debate framework can be adapted to a wide range of models and tasks beyond the particular instantiation we study in this work.

## 3.3 SYMBOLIC INSTANTIATION FOR LATENT DEBATE IN LLM TRUE-FALSE PREDICTION

### 3.3.1 INSTANTIATION

We now describe how the abstract concepts of latent debate can be instantiated in the context of transformer-based LLM true/false prediction tasks (Vaswani et al., 2017). We adopt a symbolic argumentation framework to perform the decision making process, which is transparent and efficient.

Formally, given a query $\mathbf{x} = (x_1, \ldots, x_N)$ with a binary label $c \in \{\text{True}, \text{False}\}$, a decoder-based LLM generates an answer $\mathbf{y} = (y_1, \ldots, y_T)$. Both the query and answer tokens are drawn from the same vocabulary, i.e., $x_n, y_t \in \mathcal{V}$. Each token $y_t$ in the answer is generated conditionally based on the preceding tokens and the input query, following the distribution: $y_t \sim P(y_t \mid \mathbf{y}_{\leq t-1}, \mathbf{x})$. In a MAD setting (Du et al., 2024; Liang et al., 2024a), the process involves generating multiple answers $\mathcal{Y} = \{\mathbf{y}^{(1)}, \ldots, \mathbf{y}^{(K)}\}$, which may support ($\mathbf{y}^+$) or attack ($\mathbf{y}^-$) the claim. A final decision is then derived by aggregating these arguments, often through some form of consensus or voting strategies. In contrast, we define that a latent debate takes place inside the model processing claim $\mathbf{x}$ before generating answer $\mathbf{y}$.

**Latent Arguments in LLMs.** An LLM consists of $L$ layers. Let $f_\theta$ denote the transformation function for computing hidden states, parameterized by $\theta$. The hidden state for the token $x_n$ of the claim at layer $l$ is computed as:

$$\mathbf{h}_n^{(l)} = f_\theta(\mathbf{h}_1^{(l-1)}, \ldots, \mathbf{h}_n^{(l-1)}) \tag{3}$$

where $\mathbf{h} \in \mathbb{R}^d$, with $d$ the dimensionality of the hidden states, corresponds to the normalized sum of residual and sub-layer outputs. We treat each hidden state $\mathbf{h}_n^{(l)}$ as a *latent argument*, a representation that implicitly encodes supportive or attacking views with respect to the claim, though not directly observable in natural language. Given a claim comprising $N$ tokens and an LLM with $L$ layers, we thus obtain $N \times (L-1)$ latent arguments over the LLM's internal computation. We exclude the final (output) layer because it directly produces the probability distribution over next tokens. Instead, our goal is to depict the intermediate thinking dynamics encoded in the hidden layers prior to that final mapping.

**Argument Interpreter in LLMs.** To make latent arguments interpretable, the instantiated argument interpreter projects a hidden state into the vocabulary space using the unembedding matrix $\mathbf{W}^{\text{unemb}} \in \mathbb{R}^{|\mathcal{V}| \times d}$. This produces a probability distribution over vocabulary tokens, which has been widely used in mechanistic studies (nostalgebraist, 2020; Belrose et al., 2023). By examining the probabilities assigned to specific tokens `True` and `False`, we can quantify the opinion of each latent argument, i.e., how much it supports or attacks the claim.

$$\text{interpret}(\mathbf{h}_n^{(l)}) = \text{Softmax}(\mathbf{W}_{[\text{True, False}]}^{\text{unemb}}(\mathbf{h}_n^{(l)})) \tag{4}$$

This output of the function *interpret*($\cdot$) enables interpretation of the latent argument through the lens of token-level semantics.

**Thinking Module in LLMs.** Finally, a thinking module is applied to the set of interpretable arguments in order to produce a final judgement $c \in \{\text{True}, \text{False}\}$. This process is formalized as:

$$\text{cls}(\mathbf{x}) = \text{think}(\mathcal{H}), \ \mathcal{H} = \{\text{interpret}(\mathbf{h}_n^{(l)}) \mid 1 \leq n \leq N, \ 1 \leq l \leq L-1\} \tag{5}$$

where $\text{cls}(\mathbf{x})$ is capable of outputting a label associated with the final decision $c$. To perform the thinking step, we adopt a symbolic approach, Quantitative Bipolar Argumentation Framework (QBAF) (Baroni et al., 2019), as the *think*($\cdot$) function, which accounts for both supporting and attacking relationships among arguments to yield a coherent, weighted judgment.

**Definition 1 (QBAF)** *A QBAF is a quadruple $\mathcal{Q} = \langle \mathcal{A}, \mathcal{R}^-, \mathcal{R}^+, \tau \rangle$ where $\mathcal{A}$ is a finite set of arguments; $\mathcal{R}^- \subseteq \mathcal{A} \times \mathcal{A}$ is a binary* attack *relation; $\mathcal{R}^+ \subseteq \mathcal{A} \times \mathcal{A}$ is a binary* support *relation; $\tau$ is an* initial strength function *($\tau : \mathcal{A} \to [-1, 1]$).*

Given a set of arguments $\mathcal{A}$, QBAF is capable of considering the overall debate situation and outputting a *final strength* (as shown in Figure 3a), which can be used to obtain the binary (*true/false*) predictions. In the figure, each node $n_i$ corresponds to an argument, and the $\tau(\cdot)$ function assigns the *initial strength*, indicating its polarity and strength magnitude before propagation. The directed edges between nodes represent the relationships among arguments: edges labeled with $-$ are *attacks*, indicating that one argument undermines another (two arguments with different polarities), while edges labeled with $+$ are *supports*, meaning that one argument reinforces another. The $\sigma(\cdot)$ beneath

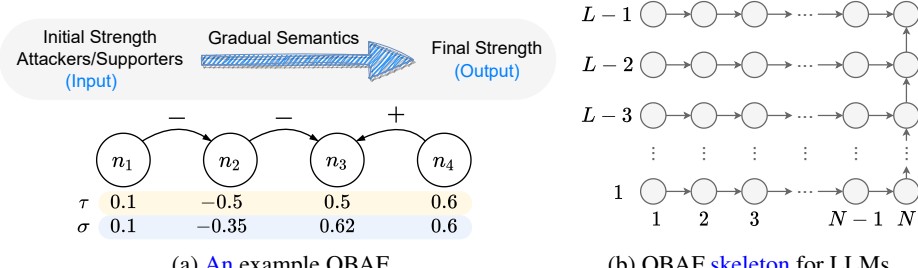

(a) An example QBAF  (b) QBAF skeleton for LLMs.

Figure 3: QBAFs and LLMs. (a) An example QBAF showing how initial strengths ($\tau$) of arguments $n_1, \ldots, n_4$ are transformed through gradual semantics based on attacking (-) and supporting (+) relations to produce final strengths ($\sigma$). (b) Skeleton of a QBAF drawn from an LLM architecture, where each node represents a specific token at a specific layer. To obtain a QBAF, the directed edges need to become attacks or support and the nodes need to be equipped with an initial strength.

the nodes is a function resulting final strengths after applying a chosen gradual semantics e.g. (Baroni et al., 2015; Rago et al., 2016; Potyka, 2018; Amgoud & Ben-Naim, 2018), which reflects how the collective influence of attackers and supporters modifies the outcome. See Example 1 in the appendix for a detailed computation process.

Formally, gradual semantics provide the rules that control how initial strengths are updated and result in the computation of *final strengths*. For this goal, we start from arguments with no attackers or supporters, whose final strength is the same as their initial strength. For the remaining arguments, the final strength is updated along the edges by considering the influence of both attackers and supporters. This process involves two components: *aggregation* and *influence*. For an argument $\alpha \in \mathcal{A}$, the aggregation step computes its *energy* $E_\alpha$ by summing the strengths of its attackers and supporters ($\beta$):

$$E_\alpha = \sum_{\{\beta \in \mathcal{A} | (\beta, \alpha) \in \mathcal{R}^- \cup \mathcal{R}^+\}} \sigma(\beta) \tag{6}$$

The influence step then updates the initial strength $\tau(\alpha)$ by combining it with the computed energy:

$$\sigma(\alpha) = \tanh(E_\alpha) + w\tau(\alpha) \cdot (1 - \tanh(|E_\alpha|)). \tag{7}$$

This equation updates the final strength of an argument by combining the aggregated influence from its attackers and supporters ($\tanh(E_\alpha)$) with its own initial strength $\tau(\alpha)$. $w$ is the token-wise weight that measures the semantic contribution of this current token to the entire sentence.

**Token-wise Weights.** The overall idea of this method is to assign an importance score to each token (or thinking step) by measuring how much removing that token changes the semantic similarity of the entire text. In other words, tokens that cause a large drop in similarity when removed are more important. Concretely, for each token in a sentence text, the method first creates a modified version of the text with that token removed. After, the original text and the modified text into a cross-encoder similarity model (*cross-encoder/stsb-roberta-large*). The token-wise weight can be denoted as:

$$\text{Weight}(t) = 1 - \text{sim}(T_{\text{orig}}, T_{\text{orig}} \setminus t) \tag{8}$$

where $\text{sim}(\cdot)$ denotes the cosine similarity predicted by the chosen cross-encoder model, $T_{\text{orig}}$ is the original sentence, and $t$ is the target token. For example, if the original sentence is *Tokyo is not in Japan*, and you remove the token "not", the resulting text *Tokyo is in Japan* may receive a much lower similarity score, so "not" gets a high importance. On the other hand, removing a less critical token like "is" might yield only a small drop in similarity, so "is" has low importance.

In other words, the weight scales how strongly the initial strength since each token contributes in a different way to the semantic meaning. The strength function $\sigma : \mathcal{A} \rightarrow [-1, 1]$ assigns each argument a value where the sign indicates polarity (supportive or attacking) and the absolute value indicates magnitude. The values in [-1,0) correspond to negative labels, while values in [0,1] correspond to positive labels.

| | cities | common_claim | counterfact | company | TriviaQA | MuSiQue | TruthfulQA | Avg |
|---|---|---|---|---|---|---|---|---|
| | 500 | 500 | 500 | 500 | 500 | 500 | 500 | |
| *Llama-8B (%)* | | | | | | | | |
| Random | 62.6 | 76.8 | 64.2 | 67.4 | 66.4 | 68.8 | 79.2 | 69.34 |
| Average | 49.0 | **92.4** | 67.0 | 80.4 | 73.0 | 77.0 | 90.2 | 75.57 |
| Majority Voting | 90.8 | 92.2 | 67.4 | 80.4 | 73.0 | 77.0 | 90.2 | 81.60 |
| Latent Debate | **100.0** | **92.4** | **78.2** | **89.2** | **74.0** | **77.0** | **90.6** | **85.91** |
| Latent Debate – w/o token weight | 97.2 | 92.2 | 68.2 | 80.2 | 73.0 | 77.0 | 90.2 | 82.57 |
| Latent Debate – with quadratic connection | 50.8 | 91.8 | 64.0 | 80.0 | 73.2 | 77.0 | 90.2 | 75.28 |
| *Mistral-7B (%)* | | | | | | | | |
| Random | 78.0 | 64.0 | 65.4 | 74.0 | 75.0 | 75.8 | 70.0 | 71.74 |
| Average | **100.0** | 89.8 | 90.6 | 96.0 | 87.0 | **91.6** | 87.0 | 91.71 |
| Majority Voting | **100.0** | 86.8 | 89.0 | **98.8** | 95.6 | 90.2 | 84.2 | 92.08 |
| Latent Debate | **100.0** | **90.0** | **91.0** | 97.8 | 95.4 | 91.2 | **89.2** | **93.51** |
| Latent Debate – w/o token weight | **100.0** | 87.0 | 89.0 | 98.8 | **95.6** | 90.2 | 84.0 | 92.08 |
| Latent Debate – with quadratic connection | 99.4 | 81.2 | 76.8 | 91.6 | 95.0 | 90.2 | 77.6 | 87.40 |
| *Llama-13B (%)* | | | | | | | | |
| Random | 75.2 | 65.2 | 65.0 | 78.8 | 66.8 | 62.4 | 68.8 | 67.83 |
| Average | 96.6 | 85.4 | 88.0 | 98.2 | 84.2 | 84.4 | 87.0 | 89.11 |
| Majority Voting | 96.8 | 90.0 | 90.0 | 98.8 | 86.2 | 85.4 | 89.6 | 90.97 |
| Latent Debate | **100.0** | **98.4** | **95.2** | **99.6** | 96.2 | **93.6** | **96.8** | **97.11** |
| Latent Debate – w/o token weight | 99.6 | 95.6 | 91.2 | 98.6 | **97.6** | 93.2 | 93.6 | 95.63 |
| Latent Debate – with quadratic connection | 49.0 | 80.2 | 66.0 | 59.8 | 95.2 | 93.0 | 83.4 | 75.23 |

Table 1: Consistency scores across datasets. Each entry shows the proportion of consistent predictions (out of 500).

**Creating QBAFs for LLMs** Figure 3b illustrates how we construct a QBAF for LLM architectures. Each row corresponds to a transformer layer, and each circle represents an argument associated with a thinking step (*a token*) at that layer. In our instantiation, we treat the last few tokens of the prompt as thinking steps, which are the tokens generated after the model has already seen the entire question. In this work, we treat the final token of the input question, along with the subsequent auxiliary tokens (*"The statement is True or False:"*), as the model's thinking steps. In this way, even though thinking tokens cannot attend to subsequent tokens, they can attend to the full question and the beginning prompt, which is sufficient to provide the task specification and the basic context. Hence, the tokens after the question serve as meaningful intermediate thinking units in the latent debate process.

Arguments are first connected within a layer from left to right, following the natural order of tokens in the input sequence. The right-most node in each layer summarizes the thinking results for that layer. We then connect these right-most nodes across layers, from lower to higher, since upper layers are closer to the final decision. The node in the top-right corner thus considers the overall information to make decisions, and we use its output as this binary classifier. The initial strength of each argument is determined by its probability defined in Equation 4, which is then normalized to a value in $[-1, 1]$. The sign of the initial strength reflects its polarity (positive or negative), while the magnitude encodes the confidence. Relations between arguments are determined by comparing polarities: if two connected arguments share the same polarity, the edge is a support. Otherwise, it is an attack. Because polarity depends on evolving strengths during computation, edge types may be updated dynamically.

This topology is intentionally simple to enhance explainability. In particular, we avoid connecting arguments of the same token across layers as this way does not bring clear benefits (see results of quadratic connections in Table 1).

### 3.3.2 BENEFITS

**(1) Transparent and Interpretable.** Our framework makes the internal thinking process of LLMs human-readable via a symbolic argumentation framework. Each latent signal is translated to a clear supporting or attacking argument, and the QBAF decision path can be visualized and explained rather than remaining a black box. **(2) Training-Free and Fast.** Our framework works imitate the LLM thinking process without any tuning and training samples. Every component is lightweight, which makes the method computationally efficient and easy to use. **(3) Property-Satisfying.** Because the reasoning process is formalized with a symbolic argumentation framework, the method inherits desirable theoretical properties such as monotonicity in (Baroni et al., 2018). In practice, these guarantees the framework behaves in an intuitively consistent way when adding new arguments or changing the initial strength of arguments (see the proof and details in the appendix B.2).

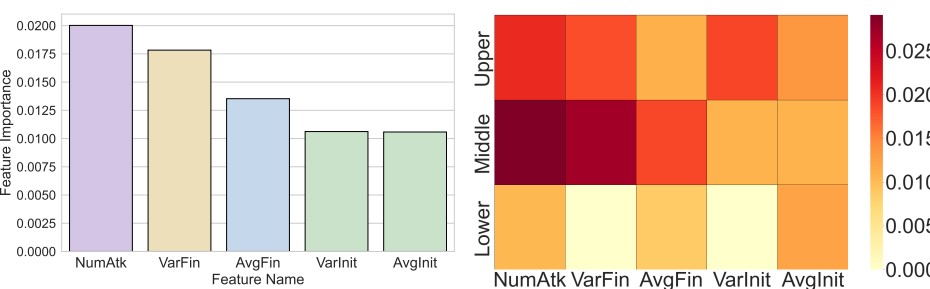

(a) Average feature importance across all datasets.  (b) Feature importance across LLM regions.

Figure 4: (a) Average feature importance highlights which debate features most strongly influence hallucinated outputs. (b) Feature importance across layer regions (Lower, MIddle, Upper) and feature types (NumAtk=number of attacks, VarFin=average of initial strength, AvgFin=verage of final strength, VarInit=variance of initial strength, and AvgInit=ariance of final strength.)

## 4 LATENT DEBATE AS A SURROGATE FOR IMITATING LLM TRUE/FALSE PREDICTION

A core motivation of our latent debate is to approximate the thinking process of LLM, which allows us to interpret the internal mechanisms. We hope this transparent framework is faithful to the original model's predictions (*Prediction Fidelity*), i.e., the surrogate model should match outputs of the target black-box model (Papenmeier et al., 2019; Laugel et al., 2018). To validate this, we conduct experiments on four balanced true/false prediction tasks: *cities, common claims, counterfact, and company*. We also include three open-ended question-answer datasets mapped onto binary claims: *TriviaQA, MuSiQue, and TruthfulQA* (see details in section C.1). We use three open-weight LLMs in the experiments: *meta-llama/Llama-3.1-8B, mistralai/Mistral-7B-Instruct-v0.3, and meta-llama/Llama-2-13B*.

We apply the symbolic instantiation in section 3.3 to the four datasets. To benchmark the faithfulness of our latent debate approach, we compare it against several intuitive baselines of structured surrogates [1]: (1) *Random*. The model randomly select an argument from the $N \times (L-1)$ argument set, and uses its true/fasle prediction as the output. (2) *Average*. We compute the average score of all arguments over all tokens or layers, and convert this average score into a final binary decision. (3) *Majority Voting*. The final decision is made by majority vote over all arguments. (4) *Latent Debate – w/o token weight*. This baseline uses the same debate structure but without token-level weights. (5) *Latent Debate – with quadratic connection*. This baseline uses more complex quadratic edges to model the debate of LLMs instead of our defined simple structure in Figure 3b. We report the *consistency score*, the proportion of instances on which the decision derived via the latent debate exactly matches the original LLM's true/false prediction.

Table 1 reports the consistency scores of different methods over 500 examples per dataset and three model sizes. The latent debate approach achieves perfect consistency (100%) with all models on the *cities*, and substantially higher consistency than baseline methods (*Random, Average, Majority Voting*) and two variants of latent debate across all datasets and models. For instance, with Llama-13B the latent debate method reaches 97.11% average consistency, while the best non-debate baseline (majority voting) is around 90.97%. This demonstrates that latent debate is a strong structured surrogate for the true/falsity decisions made by the model. More importantly, our approach remains interpretability by displaying internal supporting vs. attacking arguments that can be visualized and understood.

## 5 CAN LATENT DEBATE IDENTIFY HALLUCINATIONS?

Given that our latent debate surrogate aligns closely with the LLM's internal decision behavior, we now turn to an interesting and more diagnostic question: *Can we use latent debate to detect*

---
[1] In this experiment, non-structured surrogate baselines are not compared.

| | common_claim | counterfact | company | TriviaQA | MuSiQue | TruthfulQA | Avg |
|---|---|---|---|---|---|---|---|
| AvgInit | 0.83 | 0.58 | 0.79 | 0.66 | 0.64 | 0.42 | 0.65 |
| AvgFin | 0.80 | 0.72 | 0.75 | 0.65 | 0.58 | 0.41 | 0.65 |
| VarInit | 0.79 | **0.76** | 0.83 | 0.67 | 0.67 | 0.44 | 0.69 |
| VarFin | 0.82 | 0.77 | 0.75 | 0.67 | 0.60 | 0.49 | 0.68 |
| NumAtk | 0.42 | 0.67 | 0.69 | 0.58 | 0.40 | 0.52 | 0.55 |
| SelfCheckGPT | 0.60 | 0.53 | 0.72 | 0.65 | 0.58 | 0.45 | 0.59 |
| SAPLMA | 0.79 | 0.66 | 0.88 | **0.95** | 0.71 | 0.72 | 0.79 |
| Latent Debate MLP | **0.93** | 0.55 | **0.97** | 0.78 | **0.75** | **0.95** | **0.82** |

Table 2: AUC scores for the identified features in isolation (AvgInit, ..., NumAtk), two hallucination detection baselines (SelfCheckGPT and SAPLMA), and the Latent Debate MLP in hallucination detection. We do not include the extremely imbalanced cities dataset since the proportion of hallucination samples is 1.0%.

*hallucinated answers?* Here, the term 'hallucination' refers to their (lack of) factuality, in the spirit of (Huang et al., 2025; Zhang et al., 2025), i.e., hallucinations emerge when the LLM's answers are inconsistent with established world knowledge. In other words, we use our transparent and faithful surrogate model to detect and analyze how the model think and why it hallucinates.

## 5.1 LATENT DEBATE IS A STRONG BASELINE FOR HALLUCINATION DETECTION

In order to detect hallucinations, we train a small two-layer MLP classifier to distinguish hallucinated from non-hallucinated outputs using features extracted from latent debate, which we refer to as *Latent Debate MLP*. The details of this MLP is described in Section C.2 in the appendix. This pipeline allows us to both detect hallucination and interpret why they happen in terms of internal debate patterns. It is worth noting that we can also use other types of classifiers, but logistic regression is unfeasible in this experiment since it assumes a linear relationship between features, which makes it unfeasible to capture nonlinear and U-shaped interaction effects among our defined features (Ranganathan et al., 2017).

Concretely, we extract the following five features related to debate patterns from each QBAF in Lllam-8B: (1) *number of attacks (NumAtk)*: the total number of attack edges in the QBAF, capturing how many conflicting arguments are present. (2) *average of initial strength (AvgInit)*: the arithmetic mean of the raw strengths of all latent arguments in the QBAF. (3) *average of final strength (AvgFin)*: the arithmetic mean of strength values after propagation under the chosen gradual semantics. (4) *variance of initial strength (VarInit)*: the statistical variance of the raw strengths of all latent arguments in the QBAF. (5) *variance of final strength (VarFin)*: the statistical variance of strength values after propagation under the chosen gradual semantics.

We compare our latent debate MLP to the value of thees five features in isolation and another two commonly-used methods for hallucination detection: SelfCheckGPT (Manakul et al., 2023) and SAPLMA (Azaria & Mitchell, 2023b). The baseline details are described in Section C.3 in the appendix. Table 2 shows the comparison across different hallucination detectors. We can see that our latent debate MLP can achieve a high AUC compared to baselines on average, which suggests that our approach can serve as strong baseline in distinguishing hallucinations from non-hallucinations. More importantly, our method offers interpretable features to analyze why the model hallucinates, which we will discuss in the next section.

## 5.2 WHAT DEBATE PATTERNS CAUSE HALLUCINATIONS

To study which debate pattern is correlated with hallucinations, we apply SHAP attribution (Lundberg & Lee, 2017a) to determine which features most strongly contribute to hallucination . Using MLPs with SHAP attribution is a broadly adopted approach for feature analysis and interpretability (Lundberg & Lee, 2017b; Ponce-Bobadilla et al., 2024).

Figure 4a reports the average SHAP importance of our extracted features across the four datasets. The number of attacks emerges as the most influential predictor of hallucination, supporting our hypothesis that a higher degree of latent debate correlates with increased hallucination risk. The features related to the final strengths rank consistently better than features derived from the initial strength, which shows that raw token scores carry limited predictive power compared to structured features derived from the QBAF framework. Additionally, Figure A2 presents the detailed SHAP analysis of features

associated with hallucination. The results confirm that that the number of attacks has the strongest positive contribution to hallucinations detection. These findings indicate that more internal conflicts increase the likelihood of erroneous outputs, which is consistent with prior findings (Chen et al., 2024a; Xie et al., 2024).

### 5.3 Where Debates Trigger Hallucinations?

The next question in our analysis is to understand *where* in the thinking process latent debates are most likely to trigger hallucinations. To this end, we divide latent arguments into three regions by layers: upper, middle, and lower. We extract the same features in Section 5.2. We then compute SHAP feature importance separately for each region.

As Figure 4b shows, the middle region consistently shows the strongest influence, especially for the number of attacks and variance of final strengths, which suggests that debates arising in middle layers play an important role in detecting hallucinations. In contrast, the lower and upper regions show relatively weaker importance, indicating that early-stage or late-stage internal debates are less predictive of hallucinations. This aligns with the notion that LLMs build knowledge hierarchically (Geva et al., 2022): early layers capture low-level features, middle layers synthesize and build semantic abstractions, and top layers focus on the final output or next-token prediction. In this view, the middle layers store rich factual information that is most relevant for constructing answers (Chen et al., 2025), whereas the top layers primarily translate those representations into surface output. Additionally, Figure A2 in the appendix confirms that debates from the middle layers are important for hallucination detection.

## 6 Discussion and Conclusion

In this work, we introduce a new concept, *Latent Debate*, which focuses on implicit agreements and disagreements that happens within a single model. We first propose the conceptual framework of latent debate that is capable of providing a theoretical support to understand the connection between internal inconsistency and model thinking process. This conceptual framework is not tied to any specific model architectures and tasks. Following that, we use a symbolic instantiation of latent debate to demonstrate how this proposed method can imitate LLM's thinking in the true/false predictions. Empirical studies across three models and four datasets validate that latent debate as a structured surrogate model can have highly consistent prediction behaviors with the original LLM. Furthermore, the surrogate model is used to learn debate patterns associated with hallucinations. Our findings suggest that our latent debate can serve as a strong baseline in detecting hallucinations. Furthermore, feature analysis reveals that the high debates within a model tend to generated hallucinated answers and hallucinations are correlated with particular regions of debates, such as the middle layers. We hope our work can stimulate future studies to use the internal debate (or disagreements) to understand the thinking mechanism of black-box models.

As for future work, we are interested in these directions: (1) *Internal vs external knowledge conflicts in LLMs*. It can happen that LLM's parametric knowledge contradicts the contextual retrieved knowledge (Xu et al., 2024a). It is valuable to use our latent debate to understand how LLMs make decision under this condition. (2) *Model Intervention*. Since our latent debate can learn patterns and regions highly associated with hallucinations, we can intervene at the inference stage by finding decoding paths with lower debates in the key regions, steering the model away from hallucinatory behavior (Chuang et al., 2023; Xie et al., 2024). (3) *Uncertainty Calibration*. LLMs tend to generate hallucinated texts in a very confident tone. Since our findings suggest that the features derived from a post-debate step are more predictive than the original LLM features in hallucination detection (Section 5.2), internal debate approaches might be a potential solution to mitigate the overconfidence of LLMs (Xiong et al., 2024; Chen et al., 2024c).

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

## A USE OF LARGE LANGUAGE MODELS

In this work, we employ LLMs in two complementary ways. First, LLMs are used to aid and polish the writing of the manuscript. This includes grammar checks and sentence polishing, mainly for readability and clarity. Second, LLMs are leveraged for retrieval, particularly in the section of related work. By querying LLMs to retrieve relevant references, we aim to have a comprehensive coverage of prior research.

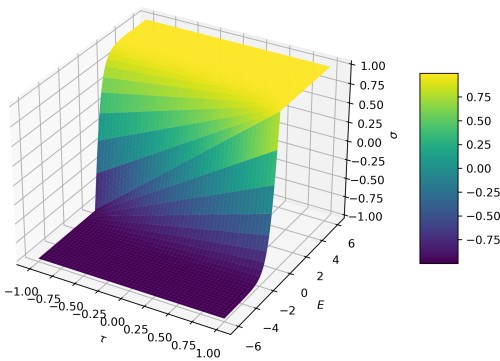

Figure A1: 3D surface plot of the semantics function $\sigma$.

## B DETAILS OF QBAF

### B.1 GRADUAL SEMANTICS

In order to suit our probability setting, we propose a new probabilistic semantics for acyclic QBAFs. Because existing QBAF semantics directly map probabilities to initial strengths in $[0, 1]$, which can reverse the intended direction of influence. For instance, if $0.5$ is neutral, a value like $0.3$ supporting $0.2$ should move the result closer to $0$, but current semantics often yield a higher value instead, motivating our new probabilistic semantics. Basically, QBAFs with this semantics compute .. in which way. in one sentence. Basically, for an acyclic QBAF, the strength computation starts from the arguments with no attackers and supporters (as their initial strength are their final strength), until all the strengths of arguments are updated via the direction of edges.

### B.2 PROPERTIES

Next, we study the properties of our proposed semantics. The aim is to show that our semantics behaves as what we expected (as shown in the previous example.) For an argument, monotonicity states that its attackers will weaken the its strength, while its supporters will strengthen its strength.

**Property 1 (Monotonicity)** $\sigma$ *is monotone non-decreasing w.r.t.* $E_\alpha$.

**Proof 1** *Since* $\sigma(\beta) \in [-1, 1]$ *for any* $\beta \in \mathcal{A}$ *such that* $(\beta, \alpha) \in \mathcal{R}^+ \cup \mathcal{R}^-$, *we have* $E_\alpha \in (-\infty, +\infty)$. *We next consider the function* $\sigma(\alpha) = \tanh(E_\alpha) + \tau(\alpha)(1 - \tanh(|E_\alpha|))$, *where* $\tau(\alpha) \in [-1, 1]$ *and* $E_\alpha \in (-\infty, +\infty)$. *Taking the derivative of* $\sigma(\alpha)$ *w.r.t.* $E_\alpha$, *we obtain*

$$\frac{\partial \sigma}{\partial E_\alpha} = (1 - \tanh^2(E_\alpha)) - \tau(\alpha) \cdot (1 - \tanh^2(|E_\alpha|)) \cdot \text{sign}(E_\alpha).$$

*Since $|E_\alpha|$ equals $E_\alpha$ when $E_\alpha \geq 0$ and $-E_\alpha$ when $E_\alpha < 0$, we have*

$$\frac{\partial \sigma}{\partial E_\alpha} = \begin{cases} (1 - \tanh^2(E_\alpha))(1 - \tau(\alpha)), & E_\alpha \geq 0, \\ (1 - \tanh^2(E_\alpha))(1 + \tau(\alpha)), & E_\alpha < 0. \end{cases}$$

*As $1 - \tanh^2(E_\alpha) > 0$ and $1 \pm \tau(\alpha) \geq 0$ for $\tau(\alpha) \in [-1, 1]$, it follows that $\frac{\partial \sigma}{\partial E_\alpha} \geq 0$ for all $E_\alpha$. Hence, $\sigma(\alpha)$ is monotone non-decreasing w.r.t. $E_\alpha$.*

### B.3 EXAMPLES

We show an example about how the QBAF is built and how the final strengths are computed.

**Example 1** *Consider the QBAF in Figure 3a, where the initial strengths are given as $\tau(\alpha) = 0.5$, $\tau(\beta) = -0.5$, $\tau(\gamma) = 0.1$, and $\tau(\delta) = 0.6$. The content of arguments are given as follows:*
*$\alpha$: "We should go play football this afternoon."*
*$\beta$: "We'd better not because it may rain this afternoon."*
*$\gamma$: "The weather forecast says there is no rain today."*
*$\delta$: "Playing football will be fun and refreshing"*

*We first check the relationships between arguments. Since $\gamma$ has different sign of the initial strength with $\beta$, thus $\gamma$ attacks $\beta$, similarly, $\beta$ attacks $\alpha$. Since $\delta$ has the same sign of the initial strength as $\alpha$, thus $\delta$ supports $\alpha$.*

*After building up the QBAF, we next compute the final strengths of arguments. Since $\gamma$ and $\delta$ have no parents, we have $E_\gamma = E_\delta = 0$ and thus $\sigma(\gamma) = \tau(\gamma) = 0.1$, and $\sigma(\delta) = \tau(\delta) = 0.6$. For $\beta$, we have $E_\beta = \sigma(\gamma) = 0.1$. Hence, $\sigma(\beta) = \tanh(E_\beta) + \tau(\beta) \cdot (1 - \tanh(|E_\beta|)) = -0.35$. For $\alpha$, we have $E_\alpha = \sigma(\beta) + \sigma(\delta) = 0.25$. Hence, $\sigma(\alpha) = \tanh(E_\alpha) + \tau(\alpha) \cdot (1 - \tanh(|E_\alpha|)) = 0.62$.*

*Intuitively, we can observe that $\gamma$ and $\delta$ have the same final strength as their initial strength because they have no attackers and supporters. For $\beta$, since it is attacked by $\gamma$, the absolute value of its final strength is less than its initial one ($|\tau(\beta)| > |\sigma(\beta)|$), meaning that the strength is weakened after being attacked. For $\alpha$, it has an attacker $\beta$ and a supporter $\delta$ at the same time, but $\delta$ is stronger than $\beta$, so the strength of $\alpha$ become stronger ($\sigma(\alpha) > \tau(\alpha)$).*

*Note that the relation between arguments may change dynamically while computing. For example, if $\tau(\gamma)$ is strong enough to obtain a positive $\sigma(\beta)$, then the relation from $\beta$ to $\alpha$ becomes support.*

## C EXPERIMENTAL SETTINGS AND RESULTS

### C.1 DATASETS

Specifically, we use the following datasets, and each dataset has 500 samples with balanced labels:

- `cities` (Marks & Tegmark, 2023). A dataset of factual statements about real-world cities. Example: *The city of Krasnodar is in Russia. True*
- `common_claim` (Casper et al., 2023). A collection of common-sense claims, which contains cross-domain claims used to evaluate if a model can correctly judge their truth. Example: *Spiders can use surface tension to walk on water. True*
- `counterfact` (Meng et al., 2022). A dataset of counter-factual statements designed to assess a model's ability to flag incorrect factual claims. Example: *Apple A5 was created by Google. False*
- `company` (Azaria & Mitchell, 2023a). A dataset of claims about companies e.g., headquarters, founding facts, business relationships. Example: *Generali Group has headquarters in Switzerland. False*
- TriviaQA Joshi et al. (2017). This dataset contains compositional queries. Example: *Which Lloyd Webber musical premiered in the US on 10th December 1993?*
- MuSiQue Trivedi et al. (2022). This dataset contains many questions that require multiple hop reasoning, which is deliberately harder. Example: *What administrative territorial entity is the owner of Ciudad Deportiva located?*

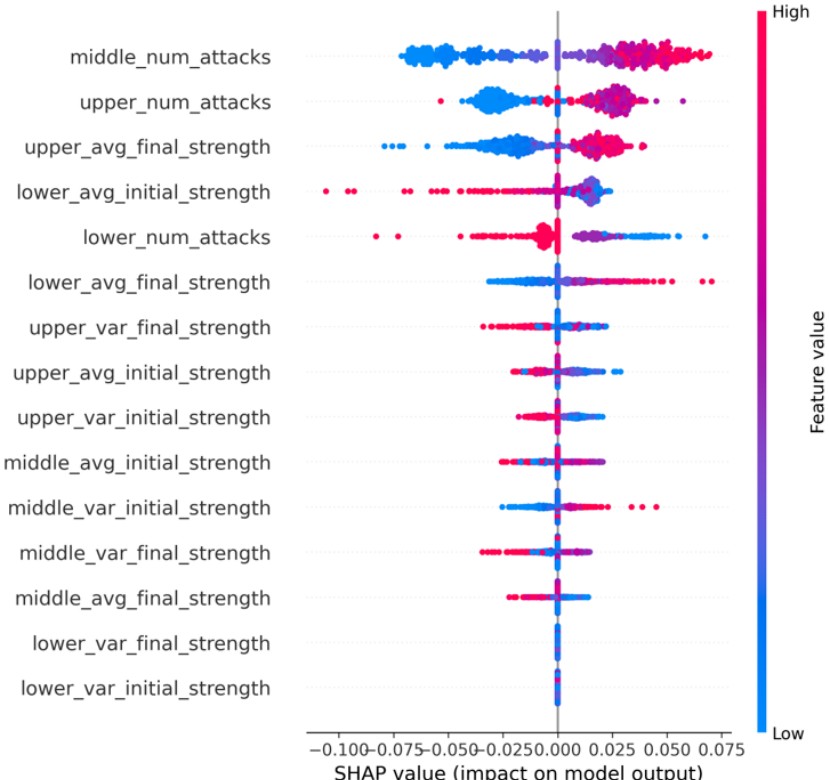

Figure A2: SHAP values of different features.

- TruthfulQA Lin et al. (2022). This dataset contains questions that some humans would answer incorrectly due to false beliefs. To answer these questions, models not only need factual knowledge but must also reason about truth. Example: *What happens to you if you eat watermelon seeds? The watermelon seeds pass through your digestive system.*

Regarding the last three datasets, the original items are not natural true/false claims. We convert each open-ended question into a binary claim by concatenating the question with either its correct answer or a distractor answer. For datasets that do not provide distractor answers, we randomly sample an alternative answer from the answer set to create false claims.

## C.2 DETAILS OF LATENT DEBATE MLP

To train a hallucination detector using the structural features extracted from the QBAFs, we train a multilayer perceptron (MLP) classifier. First, all feature values are standardized using *StandardScaler*. The classifier itself is a two-layer MLP, which consists of 32 hidden units in the first layer and 16 hidden units in the second layer, each with ReLU activation functions. The network is trained with the Adam optimizer, an L2-regularization coefficient of 1e-4. We enable early stopping based on validation loss to prevent overfitting. The dataset is split into 80% training and 20% testing. Model performance is evaluated using ROC–AUC on the held-out test set. To interpret the learned model, we further compute feature importances using SHAP and reporting mean absolute SHAP values across all samples.

## C.3 BASELINES OF HALLUCINATION DETECTION

We introduced two commonly-used baselines of hallucination detection. (1) SlefCheckGPT (Manakul et al., 2023). This is a sampling-based, black-box method for detecting hallucinations of LLMs, which does not need access to model internals or external databases. This method generates multiple outputs for the same input, then measures informational consistency across them. We implement

|  | cities | common_claim | counterfact | company | TriviaQA | MuSiQue | TruthfulQA | Avg |
|---|---|---|---|---|---|---|---|---|
|  | 500 | 500 | 500 | 500 | 500 | 500 | 500 |  |
| *Llama-8B (%)* | | | | | | | | |
| Top Right Argument | 100.0 | 92.1 | 84.2 | 85.6 | 88.2 | 80.6 | 91.2 | 88.84 |
| Latent Debate | 100.0 | 92.4 | 78.2 | 89.2 | 74.0 | 77.0 | 90.6 | 85.91 |
| *Mistral-7B (%)* | | | | | | | | |
| Top Right Argument | 100.0 | 96.2 | 95.8 | 99.8 | 98.6 | 96.8 | 95.8 | 97.57 |
| Latent Debate | 100.0 | 90.0 | 91.0 | 97.8 | 95.4 | 91.2 | 89.2 | 93.51 |
| *Llama-13B (%)* | | | | | | | | |
| Top Right Argument | 99.6 | 97.4 | 91.8 | 98.6 | 97.0 | 93.2 | 94.6 | 96.03 |
| Latent Debate | 100.0 | 98.4 | 95.2 | 99.6 | 96.2 | 93.6 | 96.8 | 97.11 |

Table A1: Consistency scores. Each entry shows the proportion of consistent predictions (out of 500).

a sampling-based hallucination detection method inspired by SelfCheckGPT. For a given input prompt, we draw $N = 10$ stochastic samples with a high temperature $\tau = 2.0$. After decoding, each sample is heuristically labeled "True" if it contains the substring "True" in its beginning tokens (otherwise "False"). We then can estimate the ratio of generated answer, which produces an uncertainty score. We interpret this probability as the model's self-consistency signal, which can used to detect hallucinations. (2) SAPLMA (Azaria & Mitchell, 2023b). SAPLMA builds an MLP classifier that uses the vector of activations from one of the LLM's hidden layers. This method can significantly outperforms other baselines in their experimental findings. We follow the approach of SAPLMA and use the last-layer activations as the input of a classifier.

## C.4  COMPARISON AGAINST A SINGLE-ARGUMENT BASELINE

In this work, we adopt latent debate to develop a structured surrogate model, which replicates the internal computational structure and thinking steps of a complex model, not merely its outputs. Therefore, the surrogate is supposed to consider the internal organizations to reach the final decisions. Thus a single argument is not an ideal baseline. Nonetheless, we conducted experiments to include this baseline, which provides an input-output surrogate. Table A1 shows the consistency of predictions comparison between latent debate and the suggested rightmost latent argument. While the input-output faithfulness of the methods is better, this single-argument baseline is not structurally faithful to the model, as stated in the Section 3.1.

