# OpenReview forum: "Latent Debate: A Surrogate Framework for Interpreting LLM Thinking"
_ICLR.cc/2026/Conference — Submitted to ICLR 2026_

### Official Review · Reviewer_1f9Y · 2025-10-28

**Soundness:** 2
**Presentation:** 1
**Contribution:** 2
**Rating:** 4
**Confidence:** 4

**Summary:**

The paper proposes Latent Debate as an interpretability framework that treats layer-level and token-level hidden states as “arguments”, maps them via an Argument Interpreter to True/False polarity and strength, and uses QBAF “thinking module” to get a final decision.

**Strengths:**

1. The overall idea sounds interesting.
2. The framework is training-free and has no intensive computational burden.
3. The authors incorporated ablation studies and some hallucination analysis on their approach.

**Weaknesses:**

1. **Evaluation looks toyish; baselines are too weak; no comparison to close related methods**

   * Datasets are toy datasets (500 each) rather than large-scale, realistic NLP benchmarks; the setting feels **too toy** to support their claims.
   * Baselines are **self-constructed and too simple**. Although the authors discussed DoLa / Logit Lens / Internal Consistency approaches briefly in text, which is good, they did not compare their method to these baselines.

2. **Writing quality is poor; core ideas are hard to follow on first pass**
   * Some keywords like "debate" and "arguments" are quite vague to understand and do not have concrete definitions. The paper probably needs a problem statement part up front (“What exactly is the concrete question you want to solve? ”). I even find this important question unclear for me on first pass.
   * Key design (token-wise weights) is moved to Appendix, which is clearly not a good practice since reviewers and readers are not required to read beyond main text.
   * Many details are missing. For instance, the crucial similarity component for token weight is not specified (which model? metric?). The appendix only states a generic “cross-encoder similarity model” without naming or detailing it.
   * There are many typos, e.g., Sec. 3.2.1 uses “trasparent” (should be transparent).
   * The illustrations are not so good as well. For example, figure 2, presented as the core framework overview, is quite ambiguous for the readers.

3. **Poor reproducibility and missing details**

   * Many key details for reproducibility are unclear, including: no specifics on the similarity backbone or metric (cosine? model name? pooling?); MLP classifier used in Section 5.1 completely lacks details on its architecture and training details.
   * To make matters worse, the authors did not provide their code. Combined with the missing details mentioned above, it is really difficult to reproduce the results, while ICLR has a high standard on reproducibility and encourages a reproducibility statement in submissions.

**Questions:**

1.  Argument Interpreter uses a Logit Lens-like approach. However, Logit Lens (and the actual logits computation in models like Llama) typically **applies LayerNorm** (or RMSNorm) after hidden states and before the `W_unemb` projection. This normalization step is critical for maintaining the correct scale and distribution of logits. Could you please explain why you chose to omit this important normalization step? Would this issue affect your framework's interpretability?

2.   All current experiments are about binary True/False tasks with clear answers. How would your framework apply to more complex tasks that lack clear binary outcomes, such as open-ended text generation or general QA?

3.  To better illustrate the working mechanism of the framework, I strongly recommend providing concrete case studies to illustrate how your framework works under concrete scenarios.

4.  Please improve the presentation quality as the current version is not so easy to follow and many necessary details for reproducibility are missing.

---

> ### Author Response · Authors · 2025-11-27
> **response (1/3)**
>
> Dear Reviewer 1f9Y,
>
> We thank you for your detailed comments and useful suggestions. We appreciate the time you invested in reviewing our paper. We are happy that you find our idea interesting and our method training-free. The manuscript has been revised according to your feedback. We hope we have addressed your concerns.
>
> **Q1: Datasets are toy datasets (500 each) rather than large-scale, realistic NLP benchmarks; the setting feels too toy to support their claims.**
>
> Thank you for your suggestions. To this end, we have conducted experiments on three complex datasets (see Section C.1).
> * TriviaQA. This dataset contains compositional queries
> * MuSiQue. This dataset contains many questions that require multiple hop reasoning, which is deliberately harder
> * TruthfulQA. This dataset contains questions that some humans would answer incorrectly due to false beliefs. To answer these questions, models not only need factual knowledge but must also reason about truth
>
> Specifically, these open-ended questions are first transformed into a binary claim using correct and distracted options. Following, we compare our latent debate to other baselines. The results of Llama-8B are shown in the table below (see all results in Table 1). The findings indicate that our latent debate has more consistent predictions with the target model.
>
> | Method                    | cities | common_claim | counterfact | company | TriviaQA | MuSiQue | TruthfulQA | Avg   |
> |---------------------------|--------|--------------|-------------|---------|----------|---------|------------|-------|
> | **Llama-8B (%)**          |        |              |             |         |          |         |            |       |
> | Random                    | 62.6   | 76.8         | 64.2        | 67.4    | 66.4     | 68.8    | 79.2       | 69.34 |
> | Average                   | 49.0   | 92.4         | 67.0        | 80.4    | 73.0     | 77.0    | 90.2       | 75.57 |
> | Majority Voting           | 90.8   | 92.2         | 67.4        | 80.4    | 73.0     | 77.0    | 90.2       | 81.60 |
> | **Latent Debate**         | **100.0** | **92.4**  | **78.2**    | **89.2** | **74.0** | **77.0** | **90.6**   | **85.91** |
> | Latent Debate – w/o token weight        | 97.2   | 92.2         | 68.2        | 80.2    | 73.0     | 77.0    | 90.2       | 82.57 |
> | Latent Debate – with quadratic connection | 50.8 | 91.8         | 64.0        | 80.0    | 73.2     | 77.0    | 90.2       | 75.28 |
>
> **Q2: Baselines are self-constructed and too simple. Although the authors discussed DoLa / Logit Lens / Internal Consistency approaches briefly in text, which is good, they did not compare their method to these baselines.**
>
> **Answer:**
>
> In this work, we adopt latent debate to develop a *structured surrogate model*, which replicates the internal computational structure and thinking steps of a complex model, not merely its outputs (please see the newly added problem statement in Section 3.1). Therefore, the surrogate is supposed to consider the internal organisations to reach the final decisions. Existing methods such as DoLa and Internal Consistency aim to improve the output accuracy, without providing a structured surrogate capable of interpreting a model’s internal reasoning. Therefore, they are not very suitable baselines for our comparison.
>
> We compare our latent debate to two variants of structured surrogates  (1) latent debate without token weight;  this uses the same debate structure but without token-level weights; (2) latent debate with quadratic connection; this baseline uses more complex quadratic edges to model LLMs instead of our defined simple structure in Figure 3b. The full results are shown in the updated Table 1.
>
> To the best of our knowledge, we are the first to propose structured surrogates for LLMs. We would be glad to include additional experiments if you have suggestions for alternative structured surrogate baselines.
>
> **Q3: Some keywords like "debate" and "arguments" are quite vague to understand and do not have concrete definitions.**
>
> > The paper probably needs a problem statement part up front (“What exactly is the concrete question you want to solve? ”). I even find this important question unclear for me on first pass.
>
> **Answer:**
>
> Thanks for your suggestion
> We have added such a problem statement in section 3.1, which provides the problem definition. The key concepts such as argument and debate are described in sections 2.3 and 3.2.
>
> **Q4: Key design (token-wise weights) is moved to Appendix, which is clearly not a good practice since reviewers and readers are not required to read beyond main text.**
>
> **Answer:**
>
> Thanks for your suggestion. We have moved the description of token-wise weights to the main content (see line 307).

---

> ### Author Response · Authors · 2025-11-27
> **response (2/3)**
>
> **Q5: Many details are missing. For instance, the crucial similarity component for token weight is not specified (which model? metric?). The appendix only states a generic “cross-encoder similarity model” without naming or detailing it.**
>
> **Answer:**
>
> Thanks for pointing this out.
> We have clarified the specific model and metric used in this work (see line 311-315).
>
> **Q6: There are many typos, e.g., Sec. 3.2.1 uses “trasparent” (should be transparent).**
>
> **Answer:**
>
> Thanks for spotting this. We have corrected this typo.
>
>
> **Q7: Many key details for reproducibility are unclear, including: no specifics on the similarity backbone or metric (cosine? model name? pooling?); MLP classifier used in Section 5.1 completely lacks details on its architecture and training details.**
>
> **Answer:**
>
> More technical details have been added in the revised manuscript, and we used blue text to highlight these updates.
> * add the details of Latent Debate MLP in section C.2
> * add the similarity backbone and metric in section 3.3.1
> * add baseline details in C.3
>
> **Q8: The illustrations are not so good as well. For example, figure 2, presented as the core framework overview, is quite ambiguous for the readers.**
>
> **Answer:**
>
> We have added more descriptions in the captions of Figures 2, 3, and 4.
>
> **Q9: To make matters worse, the authors did not provide their code.**
>
> > Combined with the missing details mentioned above, it is really difficult to reproduce the results, while ICLR has a high standard on reproducibility and encourages a reproducibility statement in submissions.
>
> **Answer:**
>
> We have uploaded the corresponding code, and we hope it can be useful for understanding the experimental details and improving the reproducibility of this work.
>
> **Q10: Argument Interpreter uses a Logit Lens-like approach.**
>
> > However, Logit Lens (and the actual logits computation in models like Llama) typically applies LayerNorm (or RMSNorm) after hidden states and before the W_unemb projection. This normalization step is critical for maintaining the correct scale and distribution of logits. Could you please explain why you chose to omit this important normalization step? Would this issue affect your framework's interpretability?
>
> **Answer:**
>
> Thanks for pointing this out. You are right. The normalization is important to scale the logit distribution. The hidden state here corresponds to the normalized sum of residual and sub-layer outputs. Since many language models use pre-norm, this output is not further normalized by a LayerNorm (or RMSNorm) after the addition. Therefore, we can project hidden states via unembeddings.
>
> To clarify this, we added one sentence in line 231.

---

> ### Author Response · Authors · 2025-11-27
> **response (3/3)**
>
> **Q11: All current experiments are about binary True/False tasks with clear answers. How would your framework apply to more complex tasks that lack clear binary outcomes, such as open-ended text generation or general QA?.**
>
> **Answer:**
>
> Thank you for your suggestions. To this end, we have conducted experiments on three complex datasets (see Section C.1).
> * TriviaQA. This dataset contains compositional queries
> * MuSiQue. This dataset contains many questions that require multiple hop reasoning, which is deliberately harder
> * TruthfulQA. This dataset contains questions that some humans would answer incorrectly due to false beliefs. To answer these questions, models not only need factual knowledge but must also reason about truth
>
> Specifically, these open-ended questions are first transformed into a binary claim using correct and distracted options. Following, we compare our latent debate to other baselines. The results of Llama-8B are shown in the table below (see all results in Table 1). The findings indicate that our latent debate has more consistent predictions with the target model.
>
> | Method                    | cities | common_claim | counterfact | company | TriviaQA | MuSiQue | TruthfulQA | Avg   |
> |---------------------------|--------|--------------|-------------|---------|----------|---------|------------|-------|
> | **Llama-8B (%)**          |        |              |             |         |          |         |            |       |
> | Random                    | 62.6   | 76.8         | 64.2        | 67.4    | 66.4     | 68.8    | 79.2       | 69.34 |
> | Average                   | 49.0   | 92.4         | 67.0        | 80.4    | 73.0     | 77.0    | 90.2       | 75.57 |
> | Majority Voting           | 90.8   | 92.2         | 67.4        | 80.4    | 73.0     | 77.0    | 90.2       | 81.60 |
> | **Latent Debate**         | **100.0** | **92.4**  | **78.2**    | **89.2** | **74.0** | **77.0** | **90.6**   | **85.91** |
> | Latent Debate – w/o token weight        | 97.2   | 92.2         | 68.2        | 80.2    | 73.0     | 77.0    | 90.2       | 82.57 |
> | Latent Debate – with quadratic connection | 50.8 | 91.8         | 64.0        | 80.0    | 73.2     | 77.0    | 90.2       | 75.28 |
>
> **Q12:To better illustrate the working mechanism of the framework, I strongly recommend providing concrete case studies to illustrate how your framework works under concrete scenarios.**
>
> **Answer:**
>
> We have provided one case study in the introduction, and we are planning to add case studies for the argumentation framework. If possible, we will add these studies to the revised manuscript before the deadline.

---

> ### Comment · Reviewer_1f9Y · 2025-11-28
>
> Thank you for clarifications and additional experiments. Unfortunately at this time the platform disables changing scores so I can not adjust it directly. However I now lean towards acceptance for AC and authors’ reference. I would be willing to raise my score to 8 if the editing permission is restored.

---

> > ### Author Response · Authors · 2025-11-28
> >
> > We thank you for raising the score from 4 to 8. We sincerely appreciate your valuable feedback, which has helped us improve the manuscript.

---

### Official Review · Reviewer_MzgQ · 2025-10-28

**Soundness:** 2
**Presentation:** 2
**Contribution:** 3
**Rating:** 4
**Confidence:** 3

**Summary:**

This paper proposes a surrogate framework for interpreting models through argumentation frameworks. The authors instantiate it using hidden states and the unembedding matrix. The results show that (1) using QBAF as the reasoning module explains model predictions better than simple feature-based methods, and (2) MLPs trained on features extracted from debate patterns predict hallucinations more accurately than using individual features.

Although I am not familiar with argumentation frameworks, I find the surrogate modeling approach conceptually interesting. The experiments demonstrate the potential of explaining LLM behaviors through the latent debate perspective. However, the writing could be clearer, and I have several questions about the methodology and experiments, which I detail below.

**Strengths:**

- The conceptual framework is intuitive and well-motivated by argumentation theory.
- While the logit lens technique used in instantiation is not novel, the experiments provide solid support for the argumentation-based approach.

**Weaknesses:**

- For the instantiation, it is unclear why all tokens are treated as thinking steps (Line 288). In LLMs, only the final token attends to all previous ones. If earlier token representations lack full contextual information, it is questionable how they can form meaningful arguments about sentence correctness.
- The writing lacks clarity. More descriptive figure captions would improve readability, rather than relying solely on definitions in the main text. The authors should also clearly define key terms (e.g., “hallucination”) in the context of the tasks.
- More experimental details are needed for reproducibility, such as the specific versions of the models used.
- The paper focuses mainly on knowledge-based toy tasks (e.g., fact judgment). It would be more valuable to examine whether reasoning questions can be explained under the same framework, which would make the work more convincing and significant.

**Questions:**

- Please specify the model versions and provide more experimental details for better reproducibility.
- Could you explain the rationale for defining each token as a thinking step? (see the first weakness)
- What proportion of the questions produce hallucinations (are hallucinations incorrect answers in this task)? Is there an obvious imbalance between hallucinated and non-hallucinated samples when training the MLP?
- Could you try to demonstrate the effectiveness of the proposed approach on reasoning tasks?

---

> ### Author Response · Authors · 2025-11-27
> **Response (1/2)**
>
> Dear Reviewer MzgQ,
>
> We thank you for your useful comments and appreciate the time you invested in reviewing our paper. We also feel encouraged by you finding our conceptual framework intuitive and well-motivated and the experiments provide solid support. The manuscript has been revised according to your feedback. We hope we have addressed your concerns.
>
> **Q1: For the instantiation, it is unclear why all tokens are treated as thinking steps (Line 288).**
>
> > In LLMs, only the final token attends to all previous ones. If earlier token representations lack full contextual information, it is questionable how they can form meaningful arguments about sentence correctness.
>
> **Answer:**
>
> In our instantiation, we treat the last few tokens of the prompt as thinking steps, which are the tokens generated after the model has already seen the entire question.
> In this work, we treat the final token of the input question, along with the subsequent auxiliary tokens ("The statement is True or False:“), as the model's thinking steps.
> In this way, even though thinking tokens cannot attend to subsequent tokens, they can attend to the full question and the beginning prompt, which is sufficient to provide the task specification and the basic context.
> Hence, the tokens after the question serve as meaningful intermediate thinking units in the latent debate process.
> We have added these descriptions in line 346.
>
> **Q2: The writing lacks clarity.**
>
> > More descriptive figure captions would improve readability, rather than relying solely on definitions in the main text. The authors should also clearly define key terms (e.g., “hallucination”) in the context of the tasks.
>
> **Answer:**
>
> We have added more descriptions in Figures 2, 3, and 4.
>
> We use the term hallucination to refer to the factuality hallucination in the spirit of [1, 2], i.e., LLM's answers are inconsistent with established world knowledge (see addition at line 436).
>
> Moreover, we have added a problem statement in Section 3.1, which provides definitions of key terms in this work.
>
> [1] Siren’s song in the ai ocean: a survey on hallucination in large language models
>
> [2]  A survey on hallucination in large language models: Principles, taxonomy,
> challenges, and open questions.
>
> **Q3:More experimental details are needed for reproducibility, such as the specific versions of the models used.**
>
> **Answer:**
>
> Thanks for your suggestions.
> * We have clarified the specific model versions in line 408
> * the technical details of latent debate MLP in Section C.2
> * baselines of hallucination detection in Section C.3
>
> We have uploaded the code for better reproducibility.
>
> **Q4:What proportion of the questions produce hallucinations (are hallucinations incorrect answers in this task)? Is there an obvious imbalance between hallucinated and non-hallucinated samples when training the MLP?**
>
> **Answer;**
>
> The proportion is shown in the table below. We can see that the dataset is moderately imbalanced on average, except for the cities dataset. At this degree of imbalance, ROC–AUC remains useful since it evaluates threshold-independent ranking performance.
>
> | Class             | cities | common_claim | counterfact | company | Avg  |
> |-------------------|--------|--------------|-------------|---------|------|
> | hallucination     | 0.01  | 0.41         | 0.16        | 0.30    | 0.24 |
> | Non hallucination | 0.99   | 0.59         | 0.84        | 0.70    | 0.76 |

---

> ### Author Response · Authors · 2025-11-27
> **Response (2/2)**
>
> **Q5: The paper focuses mainly on knowledge-based toy tasks (e.g., fact judgment).**
>
> > It would be more valuable to examine whether reasoning questions can be explained under the same framework, which would make the work more convincing and significant.
>
> **Answer:**
>
> Thank you for your suggestions. To this end, we have conducted experiments on three complex datasets (see Section C.1).
> * TriviaQA. This dataset contains compositional queries
> * MuSiQue. This dataset contains many questions that require multiple hop reasoning, which is deliberately harder
> * TruthfulQA. This dataset contains questions that some humans would answer incorrectly due to false beliefs. To answer these questions, models not only need factual knowledge but must also reason about truth
>
> Specifically, these open-ended questions are first transformed into a binary claim using correct and distracted options. Following, we compare our latent debate to other baselines. The results of Llama-8B are shown in the table below (see all results in Table 1). The findings indicate that our latent debate has more consistent predictions with the target model.
>
> | Method                    | cities | common_claim | counterfact | company | TriviaQA | MuSiQue | TruthfulQA | Avg   |
> |---------------------------|--------|--------------|-------------|---------|----------|---------|------------|-------|
> | **Llama-8B (%)**          |        |              |             |         |          |         |            |       |
> | Random                    | 62.6   | 76.8         | 64.2        | 67.4    | 66.4     | 68.8    | 79.2       | 69.34 |
> | Average                   | 49.0   | 92.4         | 67.0        | 80.4    | 73.0     | 77.0    | 90.2       | 75.57 |
> | Majority Voting           | 90.8   | 92.2         | 67.4        | 80.4    | 73.0     | 77.0    | 90.2       | 81.60 |
> | **Latent Debate**         | **100.0** | **92.4**  | **78.2**    | **89.2** | **74.0** | **77.0** | **90.6**   | **85.91** |
> | Latent Debate – w/o token weight        | 97.2   | 92.2         | 68.2        | 80.2    | 73.0     | 77.0    | 90.2       | 82.57 |
> | Latent Debate – with quadratic connection | 50.8 | 91.8         | 64.0        | 80.0    | 73.2     | 77.0    | 90.2       | 75.28 |

---

### Official Review · Reviewer_HgTy · 2025-10-30

**Soundness:** 1
**Presentation:** 3
**Contribution:** 1
**Rating:** 2
**Confidence:** 4

**Summary:**

The paper hypothesizes that models, 'internally debate' the truth of things.
A framework is proposed where model activations are taken to act as arguments for or against the matter at hand or other arguments.

The framework is instantiated by using 'logit lens'-style probing in order to quantify to what degree each activation points into the 'true' or 'false' direction. Activations toward 'true' are considered arguments in favor and 'false' are against, with the degree of alignment acting as argument strength. From there it is hypothesized that 'arguments' on each token position argue for or against the 'argument' in the next token position, and that for the last token, the 'argument' argues for or against the 'argument' in the next layer. This structure is formalized as a graph, and a semantics is proposed that allows for propagating argument strengths through this graph to arrive at a final number that is hypothesized to predict the model's final prediction (between true and false).

Finally, features of the graph are used to predict hallucinations.

**Strengths:**

The paper is clearly written, results are presented clearly, and it has helpful diagrams.

The fact that the amount of disagreement between true/false directionality within a model is predictive of hallucination is a interesting finding.

**Weaknesses:**

The results require better baselines:
(1) using only the rightmost hidden state from layer L-1, since that is the 'closest' to where the actual prediction happens.
(2) compute the consistency score for all arguments, and present the max.
Without that, I don't think the benefit is properly established: to be beneficial, the outcome of the QBAF procedure should be more predictive than the variables that go into it.

The same goes for the hallucination detection, where the results are not compared to any existing methods.

The language of debate/argument is not clearly established as metaphorical.

**Questions:**

No question.

---

> ### Author Response · Authors · 2025-11-27
> **Response (1/2)**
>
> Dear Reviewer HgTy,
>
> We thank you for your critical feedback and appreciate the time you devoted to reviewing our manuscript.  We are also happy that you find our work clearly written and our hallucination detection method interesting. The manuscript has been revised according to your feedback. We hope we have addressed your concerns.
>
> **Q1: The results require better baselines**
>
> >  (1) using only the rightmost hidden state from layer L-1, since that is the 'closest' to where the actual prediction happens. (2) compute the consistency score for all arguments, and present the max. Without that, I don't think the benefit is properly established: to be beneficial, the outcome of the QBAF procedure should be more predictive than the variables that go into it.
>
> **Answer:**
>
> Thank you for your suggestions.
>
> In this work, we adopt latent debate to develop a *structured surrogate model*, which replicates the internal computational structure and thinking steps of a complex model, not merely its outputs (please see the newly added problem statement in Section 3.1). Therefore, the surrogate is supposed to consider the internal organisations to reach the final decisions. Thus a single hidden state is not an ideal baseline. Nonetheless, we conducted experiments to include this baseline, which provides an input-output surrogate (in the terminology of the new section 3.1).  The table below shows the consistency of predictions comparison on Llama8B between latent debate and the suggested rightmost latent argument. While the input-output faithfulness of the methods is similar or even better, the new baseline is not structurally faithful to the model. We have added the results of these experiments and a discussion to the supplementary material (see all results in section C.4). The best-performing argument is not compared since this baseline requires training samples, while our method is training-free. To the best of our knowledge, we are the first to propose structured surrogates for LLMs. We would be glad to include additional experiments if you have suggestions for alternative structured surrogate baselines.
>
>
> | Method            | cities | common_claim | counterfact | company | TriviaQA | MuSiQue | TruthfulQA | Avg   |
> |-------------------|--------|--------------|-------------|---------|----------|---------|------------|-------|
> | **Mistral-7B (%)**|        |              |             |         |          |         |            |       |
> | Top Right Argument| 100.0  | 96.2         | 95.8        | 99.8    | 98.6     | 96.8    | 95.8       | 97.57 |
> | Latent Debate     | 100.0  | 90.0         | 91.0        | 97.8    | 95.4     | 91.2    | 89.2       | 93.51 |
> | **Llama-13B (%)** |        |              |             |         |          |         |            |       |
> | Top Right Argument| 99.6   | 97.4         | 91.8        | 98.6    | 97.0     | 93.2    | 94.6       | 96.03 |
> | Latent Debate     | 100.0  | 98.4         | 95.2        | 99.6    | 96.2     | 93.6    | 96.8       | 97.11 |
>
> Prompted by your comment, we identified two new structured surrogate baselines: (1)  latent debate without token weight;  this uses the same debate structure but without token-level weights; (2) latent debate with quadratic connection; this baseline uses more complex quadratic edges to model LLMs instead of our defined simple structure in Figure 3b.  The table below shows the overall comparison across 7 datasets (including three new datasets TriviaQA, MuSiQue, TruthfulQA) on Llama8B. The findings indicate that our latent debate has more consistent predictions with the target model: (see also the updated Table 1 in the paper)
>
> | Method                    | cities | common_claim | counterfact | company | TriviaQA | MuSiQue | TruthfulQA | Avg   |
> |---------------------------|--------|--------------|-------------|---------|----------|---------|------------|-------|
> | **Llama-8B (%)**          |        |              |             |         |          |         |            |       |
> | Random                    | 62.6   | 76.8         | 64.2        | 67.4    | 66.4     | 68.8    | 79.2       | 69.34 |
> | Average                   | 49.0   | 92.4         | 67.0        | 80.4    | 73.0     | 77.0    | 90.2       | 75.57 |
> | Majority Voting           | 90.8   | 92.2         | 67.4        | 80.4    | 73.0     | 77.0    | 90.2       | 81.60 |
> | **Latent Debate**         | **100.0** | **92.4**  | **78.2**    | **89.2** | **74.0** | **77.0** | **90.6**   | **85.91** |
> | Latent Debate w/o token weight        | 97.2   | 92.2         | 68.2        | 80.2    | 73.0     | 77.0    | 90.2       | 82.57 |
> | Latent Debate with quadratic connection | 50.8 | 91.8         | 64.0        | 80.0    | 73.2     | 77.0    | 90.2       | 75.28 |

---

> ### Author Response · Authors · 2025-11-27
> **Response (2/2)**
>
> **Q2: The same goes for the hallucination detection, where the results are not compared to any existing methods.**
>
> **Answer:**
>
> We added two commonly used methods for hallucination detection. (1) SelfCheckGPT [1]. It checks the consistency among multiple answers for a given query, and the degree of agreement can serve as a score to detect hallucinations. (2) SAPLMA [2]. This method detects hallucinations by training a lightweight classifier on the hidden-layer activations of a language model.
>
> The table below shows the performance of hallucination detection. We can see that our Latent Debate MLP is a very competitive baseline. At the same time, our method offers better interpretability and reveals patterns causing hallucinations, e.g., the debate locations and frequencies are correlated with hallucinations: (see also the updated Table 2 in the paper). The baseline details are described in Section C.3.
>
> | Method             | cities | common_claim | counterfact | company | Avg  |
> |--------------------|--------|--------------|-------------|---------|------|
> | AvgInit            | 0.84   | **0.83**     | 0.58        | 0.79    | 0.76 |
> | AvgFin             | **1.00** | 0.80       | 0.72        | 0.75    | 0.82 |
> | VarInit            | 0.32   | 0.79         | 0.76        | 0.83    | 0.68 |
> | VarFin             | 0.83   | 0.82         | 0.77        | 0.75    | 0.79 |
> | NumAtk             | 0.99   | 0.42         | 0.67        | 0.69    | 0.69 |
> | SelfCheckGPT       | -      | 0.60         | 0.53        | 0.72    | 0.62 |
> | SAPLMA             | 0.97   | 0.79         | 0.66        | **0.88** | 0.83 |
> | Latent Debate MLP  | **1.00** | 0.79       | **0.80**     | 0.85    | **0.86** |
>
>
> [1] SelfCheckGPT: Zero-Resource Black-Box Hallucination Detection for Generative Large Language Models
>
> [2] The Internal State of an LLM Knows When It’s Lying
>
> **Q3: The language of debate/argument is not clearly established as metaphorical.**
>
> **Answer:**
>
> You are right. Here we use the terms argument and debate in a metaphorical sense.
> These arguments correspond to a latent state inside the model, and the internal debate aggregates implicit arguments to reach a final decision. We do not claim that the model engages in human-style argumentative dialogue. We tried to clarify this in the revised paper (introduction line 51). Thanks for pointing this out.

---

### Author Response · Authors · 2025-12-01
**Rebuttal Summary**

Dear Reviewers and AC,

Thank you for your valuable comments and constructive feedback. We have conducted additional experiments and revised the manuscript accordingly. Below is a summary of the major questions and our corresponding responses.

**Q1: Evaluation looks toyish (Reviewer 1f9Y); The paper focuses mainly on knowledge-based toy tasks (e.g., fact judgment). It would be more valuable to examine whether reasoning questions can be explained under the same framework (Reviewer MzgQ)**

**Answer:**

We have included three datasets that require complex reasoning. Experimental results across three new datasets and three models confirm that our framework can consistently outperform other baselines.

Specifically, we have conducted experiments on three complex datasets (see Section C.1).
* TriviaQA. This dataset contains compositional queries
* MuSiQue. This dataset contains many questions that require multiple hop reasoning, which is deliberately harder
* TruthfulQA. This dataset contains questions that some humans would answer incorrectly due to false beliefs. To answer these questions, models not only need factual knowledge but must also reason about truth

The results of Llama-8B are shown in the table below (see all results in Table 1). The findings indicate that our latent debate has more consistent predictions with the target model.


| Method                    | cities | common_claim | counterfact | company | TriviaQA | MuSiQue | TruthfulQA | Avg   |
|---------------------------|--------|--------------|-------------|---------|----------|---------|------------|-------|
| **Llama-8B (%)**          |        |              |             |         |          |         |            |       |
| Random                    | 62.6   | 76.8         | 64.2        | 67.4    | 66.4     | 68.8    | 79.2       | 69.34 |
| Average                   | 49.0   | 92.4         | 67.0        | 80.4    | 73.0     | 77.0    | 90.2       | 75.57 |
| Majority Voting           | 90.8   | 92.2         | 67.4        | 80.4    | 73.0     | 77.0    | 90.2       | 81.60 |
| **Latent Debate**         | **100.0** | **92.4**  | **78.2**    | **89.2** | **74.0** | **77.0** | **90.6**   | **85.91** |

**Q2: The results require better baselines (Reviewer HgTy); Baselines are self-constructed and too simple (Reviewer 1f9Y)**

**Answer:**

To this end, we compare our latent debate to two variants of structured surrogates: (1) latent debate without token weight; this uses the same debate structure but without token-level weights; (2) latent debate with quadratic connection; this baseline uses more complex quadratic edges to model LLMs instead of our defined simple structure in Figure 3b. We show the results of Llama-8B below (The full results are shown in the updated Table 1). The findings indicate that our latent debate can still outperform the two variants.

To the best of our knowledge, we are the first to propose structured surrogates for LLMs (see Section 3.1). We would be glad to include additional baselines if you have suggestions for alternative structured surrogate baselines.

| Method                    | cities | common_claim | counterfact | company | TriviaQA | MuSiQue | TruthfulQA | Avg   |
|---------------------------|--------|--------------|-------------|---------|----------|---------|------------|-------|
| **Llama-8B (%)**          |        |              |             |         |          |         |            |       |
| **Latent Debate**         | **100.0** | **92.4**  | **78.2**    | **89.2** | **74.0** | **77.0** | **90.6**   | **85.91** |
| Latent Debate w/o token weight        | 97.2   | 92.2         | 68.2        | 80.2    | 73.0     | 77.0    | 90.2       | 82.57 |
| Latent Debate with quadratic connection | 50.8 | 91.8         | 64.0        | 80.0    | 73.2     | 77.0    | 90.2       | 75.28 |

**Q3: The writing lacks clarity and more experimental details are needed for reproducibility (Reviewer MzgQ); Writing quality is poor and missing details (Reviewer 1f9Y)**

**Answer:**

We have revised our manuscript and added details accordingly. The following shows the major updates
* uploaded our source code for better reproducibility
* added a problem statement in Section 3.1
* added the definition of hallucination in line 444
* added the details of Latent Debate MLP in section C.2
* added more descriptions for each figure and table

We appreciate the time and effort spent on evaluating our manuscript, and we believe the comments have helped us substantially improve the soundness and readability of our work.

---

### Meta-Review · Area_Chair_KngT · 2026-01-06

**Summary:**

The paper proposes "Latent Debate," a framework that utilizes argumentation theory to create a surrogate model for interpreting the internal thinking processes of Large Language Models (LLMs). The authors attempt to map token-level and layer-level hidden states to "arguments" to predict model outcomes and detect hallucinations. While the authors define a novel metaphorical structure, the reviews were initially critical regarding the soundness of the evaluation, the strength of baselines, and the clarity of the mechanism.

**Reviewer Concerns:**

Addressed Concerns:
- Task Scope: The authors addressed the concern that the method was only tested on toy datasets by adding experiments on TriviaQA, MuSiQue, and TruthfulQA.
- Reproducibility: The authors released their code and clarified definitions regarding "hallucination" and "thinking steps" as requested by Reviewer MzgQ and 1f9Y.

Outstanding Concerns：
- Utility and Soundness of the Framework: Reviewer HgTy (Rating 2) raised a fundamental concern: for the complex QBAF (Quantitative Bipolar Argumentation Framework) to be beneficial, it must be more predictive than the simple variables feeding into it. In the rebuttal, the authors provided a "Top Right Argument" baseline (a simple probe) as requested. Crucially, the results showed that this simple baseline often performs comparably to, or even better than, the proposed complex "Latent Debate" framework (e.g., on Mistral-7B, Top Right Argument achieved 97.57% consistency vs. Latent Debate's 93.51%). This suggests the complex "debate" graph structure adds interpretability overhead without providing a clear performance advantage or necessary fidelity over simpler methods.

- Lack of Comprehensive Baseline Comparison: While the authors argued that existing methods like DoLa are not "structured surrogates" and thus shouldn't be compared, this restricts the evaluation significantly. The failure to demonstrate superiority over established, non-surrogate interpretability baselines (beyond internal ablations) limits the assessment of the method's practical contribution. The AC agrees with Reviewer HgTy that the benefits of this specific structured surrogate are not properly established against the broader field.

**Reviewer Scores:**

- Reviewer HgTy (Original: 2): Remains 2. The rebuttal confirmed this reviewer's suspicion: the requested simple baseline ("Top Right Argument") outperformed the proposed complex framework, validating that the method's added complexity yields no performance benefit.
- Reviewer MzgQ (Original: 4): Remains 4. The authors only partially addressed this reviewer's specific doubts regarding the definition of "thinking steps" and expanded the evaluation beyond toy datasets.
- Reviewer 1f9Y (Original: 4): Explicitly improves to 8. The reviewer stated they would raise their score following the release of code and new experiments. However, the AC believes this score overlooks the critical structural redundancy revealed in the response to Reviewer HgTy.

---

### Decision · Program_Chairs · 2026-01-26

Reject